# Study on the Scale Effect of Spatial Variation in Soil Salinity Based on Geostatistics: A Case Study of Yingdaya River Irrigation Area

**Li Lu [1,2], Sheng Li [1,3,*], Rong Wu [4] and Deyou Shen [5]**

1   School of Geology and Mining Engineering, Xinjiang University, Urumqi 830046, China
2   Jiangsu Zhongwu Environmental Protection Industry Development Co., Ltd., Changzhou 213000, China
3   State Key Laboratory for Geomechanics and Deep Underground Engineering, China University of Mining and Technology, Xuzhou 221116, China
4   School of Civil Engineering, Wanjiang University of Technology, Ma'anshan 232001, China
5   Xinjiang Railway Survey and Design Institute Co., Ltd., Urumqi 830046, China
*   Correspondence: lisheng@xju.edu.cn; Tel.: +86-186-9907-1866

**Abstract:** Soil salinization seriously restricts the development of agricultural economies in arid and semi-arid areas. Mastering the spatial variability characteristics of multi-scale soil salt in irrigated areas is of great significance for the improvement and utilization of saline soil and agricultural production. The middle and lower reaches of the Yingdaya River were selected as the study area, and the irrigation area was divided into three scales: the L scale (irrigation area), the M scale (township level) and the S scale (village level). A total of 131 data sets were obtained through field investigations and sampling, and the spatial variability characteristics and scale effects of the soil salt in multi-scale irrigated areas were analyzed using classical statistics, geostatistics and nested model methods. The results showed that the average soil salinities at the L, M and S scales were 1.664%, 0.263% and 0.217%, respectively, and the coefficients of variation were 2.564, 1.312 and 0.866, respectively. The soil salinities at different scales exhibited moderate spatial correlation and anisotropic characteristics, through which, the maximum variation directions for L and M were 113° and 139°, respectively, and the maximum variation direction of the S scale was 86°. The spatial distribution of the soil salinity is affected by the scale effect, but the accuracy of spatial estimations can be effectively improved by using a multi-scale nested model for interpolation. The high-value areas of soil salt in the irrigation areas were distributed in the southeastern regions of the study area, and weakened in small areas around the high-value areas. The influence of each influencing factor on the soil salinization at different scales also differed. Except for the slope, the correlations between other influencing factors and the soil salt content gradually decreased with decreases in the scale. This study provides a concise summary of the spatial variation analysis of soil characteristic variables, and also provides a scientific basis for the formulation and implementation of salinization control programs.

**Keywords:** salinization; spatial variability; Yingdaya River irrigation area

## 1. Introduction

Soil salinization restricts the agricultural development of arid and semi-arid areas [1], seriously affecting the stability of their ecosystems and the sustainable development of their agricultural economies. Therefore, accurately mastering the spatial distribution characteristics of soil salt is of great significance in preventing and controlling the occurrence and development of soil salinization, as well as the scientific management of soil salinity [2,3].

At present, experts and scholars both at home and abroad have carried out a large amount of research on the variation characteristics of soil salt. From the perspective of the research areas, most of the research has been concentrated in coastal and plain areas,

and research on arid and semi-arid alluvial–proluvial fan areas has been relatively small-scale. For example, Emadi et al. used geostatistics to study the distribution law of the soil pH and salt in the coastal areas of northern Iran [4], Said Eljebri studied the spatial distribution characteristics of the soil salt in the Doukkala irrigation plain of Morocco [5] and Abderraouf Benslama analyzed the spatial variations in the soil salinity of the palm forest in the Algeria Plain [6]. From the perspective of the research scale, most studies on spatial variations in soil salt have been concentrated on a single scale, and a few studies have specifically discussed the spatial variations in soil salt at different scales to verify the scale effect [7–10]. For example, Cui sampled soil salt from the three scales of fields, plots and ridges to explore the multi-scale variation law of soil salt in the summer in the Yellow River Delta [11], and Ren et al. studied the soil salt variation in the agricultural area of the Hetao Plain from the field scale, canal scale and regional scale [12]. Qiao et al. studied the spatial variability of soil salt content in cotton fields by multi-scale nested sampling (4 km, 500 m and 100 m) [13]. The research on soil salinities at different scales mentioned above focused primarily on analyzing the differences in the parameters and the mechanisms of spatial structure formation, but how to use and deal with multi-scale parameters based on the scale effect is still yet to be thoroughly explored. Therefore, it is necessary to study the variation characteristics of the soil salinity at the different scales of continuous spaces, and to explore and construct a nested method for multi-scale spatial structures, accurately grasping the spatial distributions of soil salinity and allowing the spatial management and control of saline soil to reach a higher level.

This study takes the Yingdaya irrigation area as its research area, studies the spatial variations in soil salt using the three scales of L (the irrigation area), M (the township level) and S (the village level), proposes a multi-scale nested model for the research process and discusses the inter-relationships and action intensities of various influencing factors and the soil salinization at the different scales, so as to effectively elucidate the spatial distribution characteristics and formation causes of the presence of soil salt in the irrigation area, providing a scientific basis for the formulation and implementation of a salinization control plan.

## 2. Materials and Methods

### 2.1. Overview of the Study Area

The Yingdaya River irrigation area is located at the south foot of the Tianshan Mountains and the middle part of the northern foot of the Tarim Basin (location: southeast of Kuqa County, Aksu Prefecture, Xinjiang, China). It is part of the alluvial plain of the Kuqa River, is mainly an agricultural area, featuring an oasis, desert and other landscapes, and is at an altitude of 950–1030 m [14]. The administrative region is subordinate to Kuqa County, which includes Arahag Town, Qiman Town, the Akwusitang Township and the Hanikatamu Township, with geographical coordinates of 82°40′~83°16′ E and 41°14′~41°37′ N. The irrigation area is approximately 50 km long from north to south, and approximately 19 km wide from east to west, encompassing a total area of approximately 706 km$^2$. The irrigated area belongs to a warm temperate continental arid climate, with dry and cold winters and hot summers. The annual average temperature is 10.1–12.8 °C, and the evaporation precipitation ratio can reach 26.4 [15]. The irrigation area is densely covered with rivers and canals, among which, the Yingdaya River and Weigan River are the main sources of irrigation water in the territory. Due to long-term dependence on Weigan River for irrigation, the river water has been reduced and even cut off at times. The main crops are wheat, corn, cotton, walnut, etc., and the dominant vegetation in the desert area is halostachys caspica, kalidium foliatum, halocnemum strobilaceum, tamarix and Populus euphratica [16]. The location of the study area is shown in Figure 1.

The study area is in the alluvial–proluvial fan, so the spatial distribution of soil types is closely related to the landform. From the northeast to the southeast of the study area, the soil particles show a sorting effect from coarse to fine. Therefore, the soil in the northeast part of the study area is mainly sand and medium coarse sand, and the degree of salinization

is poorly developed. In the southeast, the terrain is relatively flat, the soil particles are fine, the sediment is composed of sand, subsandy soil, subclay and other light loam and the degree of salinization is heavy. The study area belongs to "irrigated agriculture", in which, Yingdaya River and Weigan River are the main sources of irrigation water in the territory. Therefore, the distribution of salinization is closely related to agricultural irrigation activities. In recent years, with the rapid expansion of agricultural production scale, groundwater and soil water exchange frequently, resulting in the elevation in water level, which also provides conditions for the occurrence of salinization. The salinization landscape is shown in Figure 2.

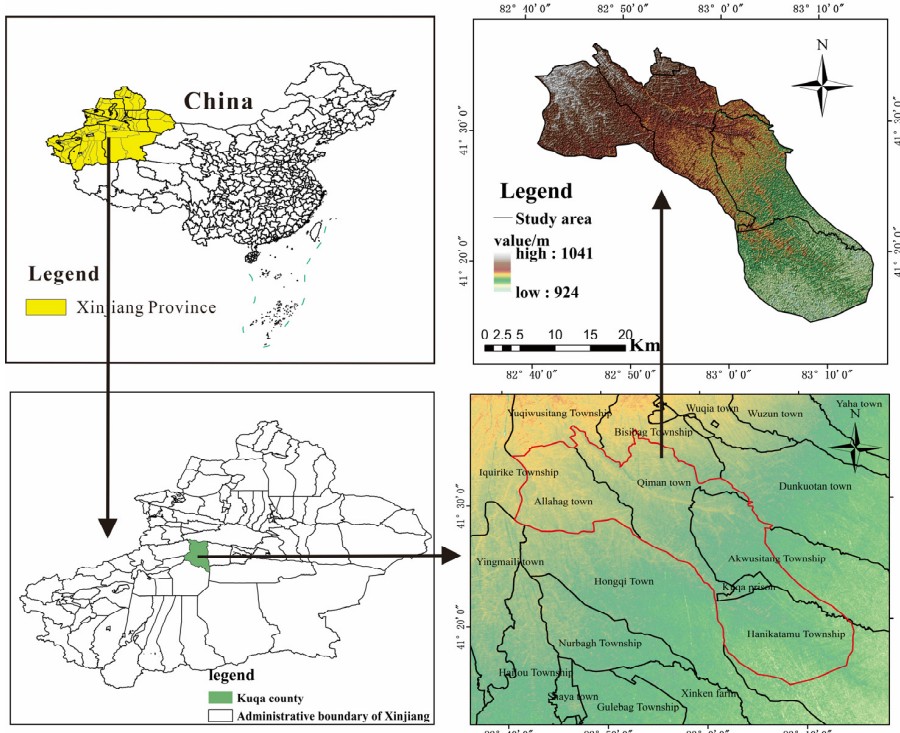

**Figure 1.** Location map of the study area.

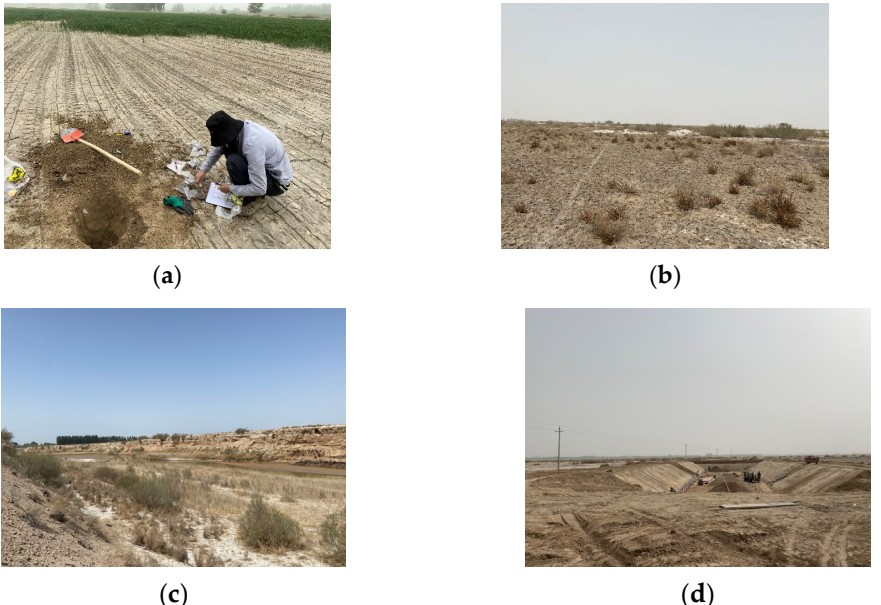

**Figure 2.** Salinization landscape in irrigated area: salinization of cultivated land (**a**); oasis edge salinization (**b**); river salinization (**c**); reclamation of saline alkali land (**d**).

## 2.2. Collection of Soil Sample

The spatial scale mainly includes the sampling amplitude and sampling granularity [17]. In this study, the sampling amplitude and grain size were changed simultaneously to represent the changes in the spatial scale, so as to facilitate the study of the spatial variability in the soil salinity at different scales. According to the administrative division principle [18], the sampling amplitudes are divided into three classifications, namely the L scale (the irrigation area), the M scale (the township level) and the S scale (the village level). The whole irrigation area is taken as the L scale, Qiman Town as the M scale and Dabozi Village in Qiman Town as the S scale. Based on the preliminary investigation and literature review for the sample area, it is generally believed that the formation process of the spatial variation in the soil salinity is significantly affected by factors such as the topography, groundwater environment, rivers, soil parent materials and cultivation management [15,19,20]. Among them, the influences of the topography and groundwater environment are generally within a larger scale of 4 km or more, which shows obvious differences. Generally, the influence of the river, soil texture and land use type can only highlight differences at scales of 2 km or above. Differences caused by farmers' personal characteristics, such as farming management and micro-topographical changes, usually occur at scales of less than 600 m [21,22]. At the same time, in order to choose the appropriate step length for the calculation and analysis of the variogram, it is necessary to sample according to the regular grid. Therefore, based on the basic principles of geostatistics on the sampling point setting and the actual situation of the study area, the sampling intervals for this study were determined to be 4.4 km, 2 km and 550 m.

Soil collection was carried out from 20 April 2021 to 5 May 2021, and a total of 131 sampling points were arranged in the study area via grid nested encryption using GIS. The L-scale sampling distance was 4.4 km, and there were 44 sampling points. The M scale featured encrypted sampling with a sampling interval of 2 km and 51 sampling points. Finally, the S scale was locally encrypted again in the same way, with a sampling interval of 550 m and 36 sampling points. The above sampling levels together constituted an interconnected nested system, as shown in Figure 3. In order to reduce soil differences, five sampling points were arranged. with the sampling point as the center and a radius of 10 m forming a plum blossom distribution, and the sampling depths were 0–30 cm. The corresponding layers of soil from five sampling points were mixed and approximately 0.5 kg of soil samples were taken by the quartering method. After sampling, the samples were numbered and sealed, and the sealed soil samples were brought back to the room for finishing, at which point, the soluble salt of the soil was measured according to the design requirements.

## 2.3. Sample Data Processing

From 10 May to 23 May 2021, the collected soil samples were brought to the laboratory, airdried at room temperature, passed from a 2 mm sieve and subjected to the determination of the salt content of the samples according to the methods for agricultural chemical analysis of soil [23]. Method: mixing with a 5:1 water–soil ratio, after shaking for 3 min, place the filtered solution to be measured in a porcelain evaporation dish with a Buchner funnel and steam it dry in a water bath to obtain residue. The organic matter in the residue was removed with hydrogen peroxide, and, finally, the porcelain evaporating dish was subjected to an oven for drying, and was weighed after cooling; the reading was employed in the formula to obtain the total amount of soil soluble salt. The descriptive statistical analysis of test samples was completed by SPSS. The semi-variogram calculation and theoretical model selection were carried out by using GS+. The spatial structure multi-scale nesting and Kriging interpolation were carried out by using the Geostatistical Analyst module in ArcGIS. The corresponding maps were drawn by using Origin and CDR. For principles and methods of geostatistics, refer to relevant literature [4,5,10].

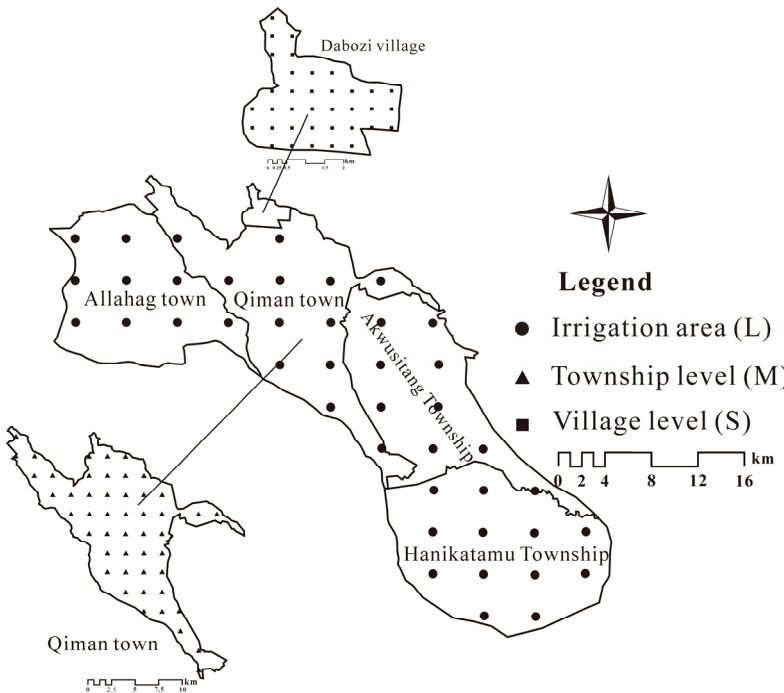

**Figure 3.** Sampling point distribution.

According to the Classification Standard of the Soil Salinization in Arid and Semi-arid Areas of China [24], the soil salt content was divided into five levels, as shown in Table 1.

**Table 1.** Classification standard of soil salinization degree in China.

| Salt Series and Zone of Application | | Soil Salt Content/% | | | | |
|---|---|---|---|---|---|---|
| | | Non | Mild | Moderate | Severe | Saltierra |
| I | Coastal, Semi-humid, Semiarid, Arid area | <0.1 | 0.1–0.2 | 0.2–0.4 | 0.4–0.6 | >0.6 |
| II | Semi-desert and Desert area | <0.2 | 0.2–0.3 | 0.3–0.5 | 0.5–1 | >1 |

*2.4. Data Analysis and Methods*

(1) Variogram and Theoretical model

As a basic function to study the spatial variation in soil attributes, some important parameters such as range (*a*), nugget($C_0$), sill($C_0 + C$) and fractal dimension can reflect the variation degree and autocorrelation range of soil characteristics on a certain scale. The calculation formula is shown in Equation (1).

$$\gamma(h) = \frac{1}{2N(h)} \sum_{i=1}^{N(h)} [Z(x_i) - Z(x_i + h)]^2 \tag{1}$$

where $\gamma(h)$ is the experimental variogram; $N(h)$ is the number of sample pairs; *h* is the vector between the two sample points; $Z(x_i)$ and $Z(x_i + h)$ are the observed values of $Z(x_i)$ at spatial positions $x_i$ and $x_i + h$, respectively.

The spatial variation causing soil salinity includes two parts: one is the nugget ($C_0$) that represents the variation caused by random factors, including sampling error and human factors such as land use mode, cultivation and irrigation within the minimum sampling interval. The other is the partial sill (C), which represents the variation caused by structural factors, mainly including natural factors such as topography, groundwater environment and soil parent material [14,20]. Nugget coefficient [$C_0/(C_0 + C)$] indicates the proportion of variation caused by random factors in the total variation in the system,

and the ratios <25%, 25~75% and >75%, respectively, indicate that the correlation of spatial variables is strong, moderate and weak. The higher the ratio, the more that soil salt variation is caused by random factors. On the contrary, salt variation is caused to a greater extent due to structural factors [25,26].

(2) Anisotropic nesting structure

For most geometric anisotropy, the directional variation diagram approximates an ellipse. At this time, let $a_1$ be the major axis of the ellipse, which coincides with the horizontal direction, $a_2$ be the minor axis, which coincides with the vertical direction (Figure 4) and k = $a_1/a_2$; then, the linear transformation matrix is A = $\begin{bmatrix} 1 & 0 \\ 0 & k \end{bmatrix}$, and the transformed $h'$ is shown by Equation (2).

$$h' = \begin{bmatrix} h'_w \\ h'_v \end{bmatrix} = \begin{bmatrix} 1 & 0 \\ 0 & k \end{bmatrix} = \begin{bmatrix} h_w \\ h_v \end{bmatrix} = \begin{bmatrix} h_w \\ kh_v \end{bmatrix} \tag{2}$$

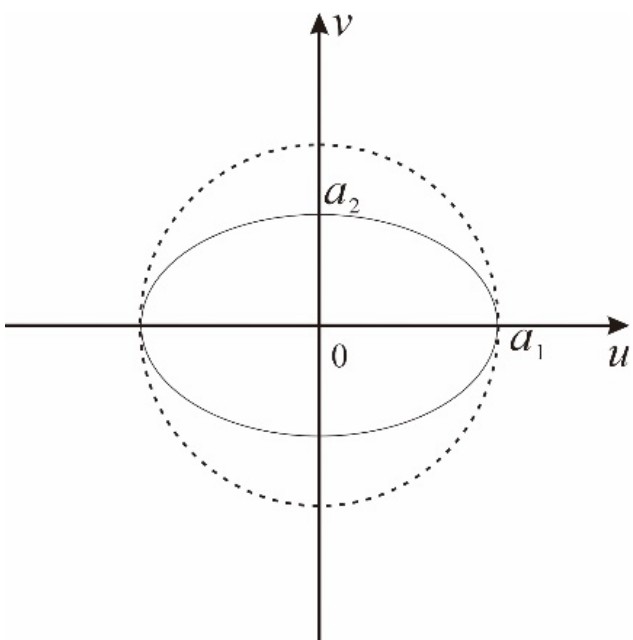

**Figure 4.** Ellipse stretching transformation.

At this time, the pattern of the shape of the ellipse becomes a circle, with the major axis as the radius (Figure 4), and the anisotropy becomes isotropy. The variation in the vertical direction is $\gamma(h')$, the horizontal variation is $\gamma(h)$ and the final anisotropic nesting result is expressed in Equation (3).

$$\gamma^*(h) = \gamma(h') + \gamma(h) \tag{3}$$

In practical research, there is no specific variation direction (Figure 5), which is generally studied by first rotating the coordinate axes by an angle $\varphi$ so that they are parallel to the major axis of the ellipse, and then converted into isotropy through a linear transformation in Equation (2).

(3) Multiscale nested structure.

Since the variation characteristics of regionalized variables on different scales are different, they cannot be represented by a simple theoretical model, but need to be described by two or more theoretical models whose structure is the superposition of multiple structures on each other, called a nested structure, which can be expressed by the sum of multiple semi variograms reflecting the changes in different scales, as shown in Equation (4).

$$\gamma(h) = \gamma_0(h) + \gamma_1(h) + \cdots + \gamma_n(h) = \sum_{i=0}^{n} \gamma_i(h) \tag{4}$$

where $\gamma_0(h)$ is the variation on the micro-scale, which is usually expressed as the spatial structure and randomness that cannot be expressed on the minimum sampling scale, which is the nugget of the nested structure; $\gamma_i(h)$ can be the same or different theoretical models, representing the spatial structure with good integration and independence on different spatial scales.

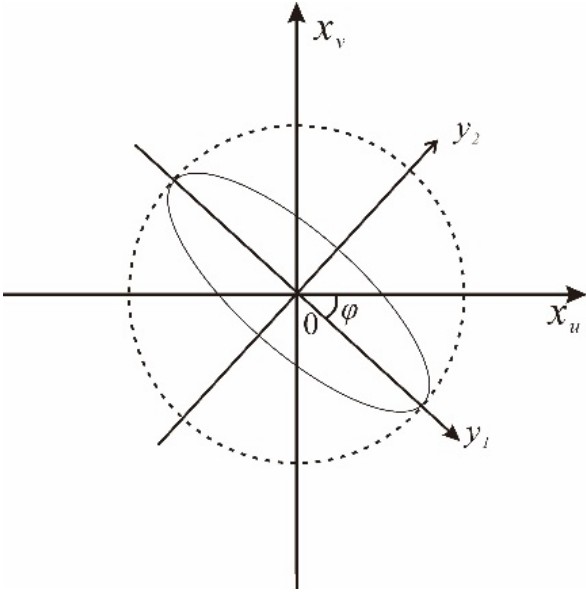

**Figure 5.** Coordinate axis rotation transformation.

(4) Model optimality test method

*I*-value test method and cross-validation method are used to test the optimality of the theoretical semi-variogram model, which is the calculation method as follows:

$$I = \overline{[\hat{Z}(x_i) - Z(x_i)]^2} \times \left[ P \times \left| 1 - \frac{1}{\left[ \frac{\hat{Z}(x_i) - Z(x_i)}{\hat{\sigma}(x_i)} \right]^2} \right| + (1 - P) \right] \tag{5}$$

$$P = \begin{cases} 0.1, & 0 \leq \overline{[\hat{Z}(x_i) - Z(x_i)]^2} \leq 100 \\ 0.2, & 100 \leq \overline{[\hat{Z}(x_i) - Z(x_i)]^2} \end{cases} \tag{6}$$

where $\hat{\sigma}(x_i)$ is the standard error of Kriging prediction, $\hat{Z}(x_i)$ is the predicted value of the $i$th sample at position $x$, $Z(x_i)$ is the measured value of the $i$th sample at position $x$ and P is an empirical parameter. The smaller the value of $I$, the better the representativeness of the theoretical model to the spatial variation structure of the soil.

(5) Grey Relational Analysis

Grey relational analysis is a multi-factorial statistical analysis method that calculates the similarity between the reference sequence and the comparison sequence; the more similar the two curves are, the closer the relation between the sequence will be. The specific methods are as follows [27]:

① Original data transformation. The selected indicators are different in physical meanings and dimensions, and we should thus adopt the method of removing dimension before comparing each data column.

② Correlation coefficient computations. It is necessary to determine the correlation coefficient $\xi_i(k)$ in each sub-sequence $x_i(k)$ and parent sequence $x_0(k)$. The computational formula of correlation coefficient in the grey system is as follows:

$$\xi_i(k) = \frac{\underset{i}{min}\,\underset{k}{min}|x_0(k) - x_i(k)| + \rho\,\underset{i}{maxmax}_k|x_0(k) - x_i(k)|}{|x_0(k) - x_i(k)| + \rho\,\underset{i}{maxmax}_k|x_0(k) - x_i(k)|} \tag{7}$$

among which:

$k = 0, 1, 2, 3, \cdots, n$, $i = 0, 1, 2, \cdots 7$, and $\xi_i(k)$ is the correlation coefficient of the data series of $x_i$ and $x_0$ at position $k$. The effect of $\rho \in [0, 1]$, which is called the resolution ratio, is to highlight the difference between the correlation coefficients. Generally, the resolution ratio is 0.5.

③ Solving the correlation degree, $r_i$. The correlation degree of the two sequences is provided by the average value of the correlation coefficient between the sub-sequence and the parent sequence at each time; that is:

$$r_i = \overline{\xi_i(k)}\frac{1}{n}\sum_{k=1}^{N}\xi_i(k) \tag{8}$$

where $r_i$ is the correlation degree between the two sequences and $N$ is the number of each sub-sequence.

## 3. Results

### 3.1. Probabilistic Statistical Analysis of Soil Salinity

Before the geostatistical analysis of spatial sampling point data, a descriptive statistical analysis should be carried out on the data based on Fisher's random statistics theory through SPSS, which can summarize the overall status of soil salinity from the statistical level.

The descriptive statistical results of the salt content of the shallow soil at the three sampling scales of the irrigation area, the township level and the village level are shown in Table 2. The average salt contents of the soil at each scale in the study area were 1.664%, 0.263% and 0.217%, respectively. According to the salinization classification standards introduced in Table 1, scale L corresponds to saltierra, and the M and S scales correspond to moderate salinization. Through the K-S test, it was found that the $p$ values were less than 0.05, which does not conform to a normal distribution. Therefore, a logarithmic transformation of the measured sample data was necessary to eliminate proportional effects. After conversion, the $p$ values of different scales were greater than 0.05, or approximately 0.05, which conformed to the normal distribution and was suitable for geostatistical analysis [28].

**Table 2.** Descriptive statistical results of soil total salinity at different scales.

| Scale | Min (%) | Max (%) | Mean (%) | SD | Skewness | Kurtosis | CV | Converted *p*-Value |
|-------|---------|---------|----------|-----|----------|----------|-----|---------------------|
| L | 0.069 | 22.636 | 1.664 | 4.266 | 3.780 | 15.315 | 2.564 | 0.200 |
| M | 0.092 | 1.570 | 0.263 | 0.345 | 3.194 | 9.909 | 1.312 | 0.087 |
| S | 0.078 | 0.618 | 0.217 | 0.188 | 1.481 | 0.471 | 0.866 | 0.047 |

In addition, with the decrease in the scale, it was found that the variation range of the soil salinity coefficients of the variation and salt content gradually decreased, through which, the coefficient of variation in soil salinity at the L scale was found to be 2.564 (strong variability), the coefficient of variation at the M scale was 1.312 (strong variability) and the coefficient of variation at the S scale was 0.866 (moderate variability). The data above show that the soil salinity in small ranges tends to be homogenized as a whole, and the soil salinity in large ranges has the characteristics of heterogeneity and complexity. There

are two possible reasons for this: one is that the soil salt content is affected by different random and structural factors at different scales; the other is the choice of the small and medium-scale study locations [15,29]. Specifically, there is a great consistency between farming and management at the village level, which inevitably leads to a more uniform distribution of soil salt and a low coefficient of variation. With the increase in the scale, especially at the irrigation district scale, and the differences in farming and management measures, land use methods and landform and groundwater environments among different towns and townships, the variation range of and variability in the soil salt content increased. The selection of the small and medium-sized study areas in the large-scale regions also affected the variation range of and variability in the soil salt. If a small-scale study area is selected in a large-scale high-value region, the small-scale salt content and variability may increase, and vice versa.

### 3.2. Spatial Scale Effect of Soil Salinity

Classical statistics can only reflect the overall changing characteristics of soil salt from the statistical significance, and cannot quantitatively describe the randomness, structure, independence and correlation of soil salt. Therefore, it is necessary to use geostatistics to analyze and discuss its spatial variation structure.

#### 3.2.1. Isotropic Analysis

After logarithmic transformation, while ignoring the variation directionality, Gs+ and the geostatistical analyst module in ArcGIS were used to perform geostatistical analysis on data at different scales, and the theoretical model and parameters of the soil salt variogram at three scales were obtained. The larger the determination coefficient ($R^2$), the smaller the residual error (RSS), which indicates that the theoretical model has a higher degree of fitting to the experimental variogram [20] (the results are presented in Figure 6 and Table 3).

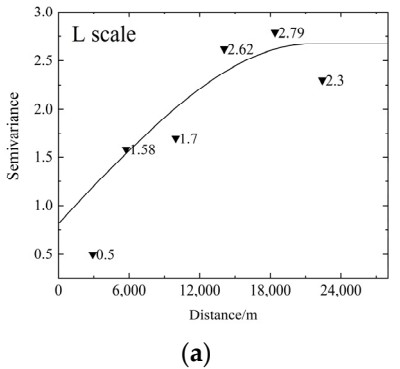 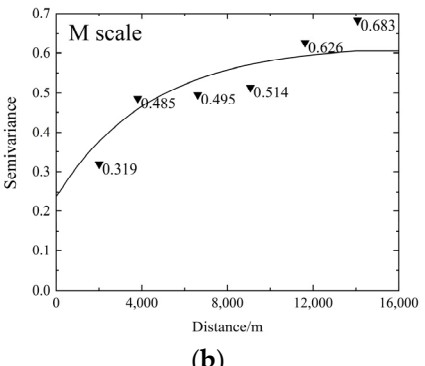 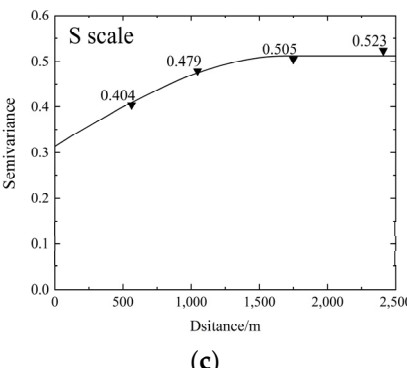

(a)  (b)  (c)

**Figure 6.** Fitting diagram of semi-variogram of soil salinity at different scales: L-scale semi-variogram fitting diagram (**a**); M-scale semi-variogram fitting diagram (**b**); S-scale semi-variogram fitting diagram (**c**).

**Table 3.** Isotropic variogram model and parameters of soil salinity at different scales.

| Project | Scale | Model | Nugget<br>$C_0$ | Sill<br>$C_0 + C$ | Range<br>$a$/km | Nugget Coefficients<br>$C_0/C_0 + C$ | $R^2$ | RSS |
|---|---|---|---|---|---|---|---|---|
| | L | Spherical | 0.801 | 2.685 | 21.460 | 0.298 | 0.394 | 2.2712 |
| Salt content | M | Exponential | 0.237 | 0.611 | 13.310 | 0.387 | 0.502 | 0.0463 |
| | S | Spherical | 0.314 | 0.512 | 1.685 | 0.613 | 0.922 | 0.0548 |

From Table 3 it can be observed that a spherical model can better express the spatial variation structure of the soil salinity on the L and S scales. On the M scale, its spatial variation structure was more consistent with an exponential model. The nugget ($C_0$), sill

(C$_0$ + C) and range (*a*) of soil salt at the different scales were quite different, and, among the nugget coefficients of soil salinity, which were 0.613, 0.387 and 0.298, respectively, all were in line with moderate spatial correlation, indicating that the soil salt variability was caused by both structural and human factors. With the increase in the sampling scale, the nugget coefficients gradually decreased, indicating that the spatial correlation gradually increased, the influence of structural factors gradually increased and the influence of human factors on the soil salinity weakened. Compared with the M and S scales, the nugget increased significantly in the L scale, indicating that the variability caused by land management measures smaller than the minimum sampling interval cannot be ignored at large scales [19]. Compared with the increases in the nugget, the increases in the sill were more significant, indicating that the variability caused by structural factors was scale-dependent, and the influencing factors that fail to reflect differences in the M and S were manifested in the L scale as the scale increases.

In general, from the L scale to S scale, the spatial autocorrelation of soil salinity is weakened, the nugget effect is increased and smaller structural features masked by the larger scales are highlighted at the smaller scales. Additionally, the R$^2$ gradually decreased with increases in the sampling spacing, indicating that the more that small-scale structural features are masked, the more the expressions of large-scale spatial structural features via variogram are gradually weakened.

The range (*a*) reflects the autocorrelation range of the variables [30], in which, the L-scale soil salinity range had the largest range at 21.46 km, the M-scale range was 13.31 km and the S-scale range was the smallest at 1.685 km. Among them, the structural factors, such as the soil parent material, the groundwater environment, the topography and the geomorphology, were relatively consistent on the S scale. Therefore, under long-term cultivation and management, the range (*a*) of the soil salinity was reduced and the distribution was relatively homogeneous. The M and L scales were mainly affected by structural factors, and the degree of influence increased with an increasing scale; specifically, the range (*a*) became significantly larger in the M scale. The results above show that the influencing factors or degrees of soil salt variability at different scales were also different. In combination with Figure 6, it can be found that the S-scale variogram image was insensitive to the change in distance, and its variation degree did not change significantly with an increase in the range, so the random factor was the dominant factor; the curvature of the L-scale and M-scale variogram image were large, and the variogram curve presented a rapid rise within their respective ranges, indicating that the structural factors were the main reason for the significant increase in the variability in salt content.

### 3.2.2. Anisotropic Analysis

In practical research, due to differences in the soil-forming environments, groundwater characteristics, topography and geomorphology in different directions and human management, the spatial structures of soil properties have a certain directionality [31]. In the analysis of variation structures, it is imperative to analyze the variation directions and to determine the maximum variation direction [18] in order to provide parameters for the construction of the nested model.

Based on this, the anisotropic parameters were obtained by using the automatic search function of the anisotropy axis in the geostatistics module. The anisotropic parameters at different scales are presented in Table 4.

It can be seen from Table 4 that, on the L scale, the soil salinity exhibited the same sill with different ranges (*a*) in the 113° and 203° directions, which corresponds to a typical geometric anisotropy. The maximum anisotropy ratio at 113° was 1.748, indicating the main direction of variation in the L scale was northwest-southeast, and this was also basically consistent with the direction of the groundwater flow field and topographic changes in the region.

**Table 4.** Fitting parameters of soil salinity anisotropy at different scales.

| Scale | Major Axis Direction Angle/° | Major Axis Range *a*/km | Minor Axis Range *a*/km | Anisotropy Ratio/k | Nugget $C_0$ | Sill $C_0 + C$ |
|---|---|---|---|---|---|---|
| L | 113 | 35.43 | 20.27 | 1.748 | 0.794 | 2.654 |
| | 315 | 23.08 | 18.44 | 1.252 | 1.21 | 2.627 |
| M | 139 | 16.91 | 5.718 | 2.957 | 0.0164 | 0.653 |
| | 49 | 5.718 | 16.91 | 0.338 | 0.0164 | 0.653 |
| | 348 | 16.89 | 10.17 | 1.661 | 0.252 | 0.602 |
| S | 86 | 3.46 | 1.527 | 2.266 | 0.223 | 0.562 |
| | 176 | 1.527 | 3.46 | 0.441 | 0.223 | 0.562 |

On the M scale, the sill of the soil salinity in the 139° and 49° directions was the same with different ranges (*a*), which corresponds to geometric anisotropy. The maximum anisotropy ratio in the 139° direction was 2.957, indicating that the most significant variation direction was 139°, which was the same as the flow direction and groundwater flow field of the Yingdaya River.

The S-scale maximum variation direction was 86° (east-west direction), and the minimum variation direction was 176° (north-south direction). Due to the small S-scale range, the influence of large-scale structural factors such as the soil-forming process and the groundwater environment on all directions on the soil salinity was the same. Therefore, the anisotropy of the soil salinity on this scale was mainly caused by differences in long-term cultivation and irrigation measures [32]. Through remote sensing images (Figure 7) and field visits, it was found that, except for the farmland land divisions and cultivation directions distributed on the west side of provincial highway S210 being slightly irregular, the land division directions and cultivation directions of other farmers were north-south and east-west, respectively. Farmland soil salinity dynamically responds to farmland management measures [32,33], with the east-west direction on the east side of provincial highway S210 being the cultivation direction of the farmers, and the same cultivation and irrigation measures in the parallel zone led to a large similarity range (large range) for the soil salinity and to the variability in a large range being reduced. The farming management measures of different farmers in the north-south direction on the east side of provincial highway S210 inevitably resulted in a small range of soil salt autocorrelation and an increased variability in a small range.

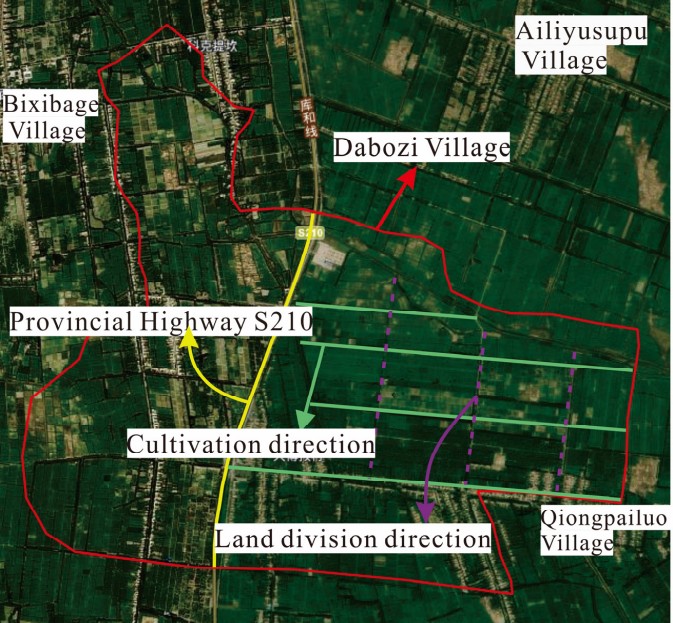

**Figure 7.** Schematic diagram of the S-scale anisotropy.

*3.3. Spatial Distribution Characteristics of Salt Content in Soil and Multi-Scale Nesting*

3.3.1. Spatial Distribution Characteristics of Soil Salinity at Different Scales

According to the optimal semi-variogram obtained above, the ordinary Kriging method was used to interpolate the spatial distribution map of soil salinity at three scales and calculate the area and proportion of different degrees of salinization (results presented in Figure 8 and Table 5).

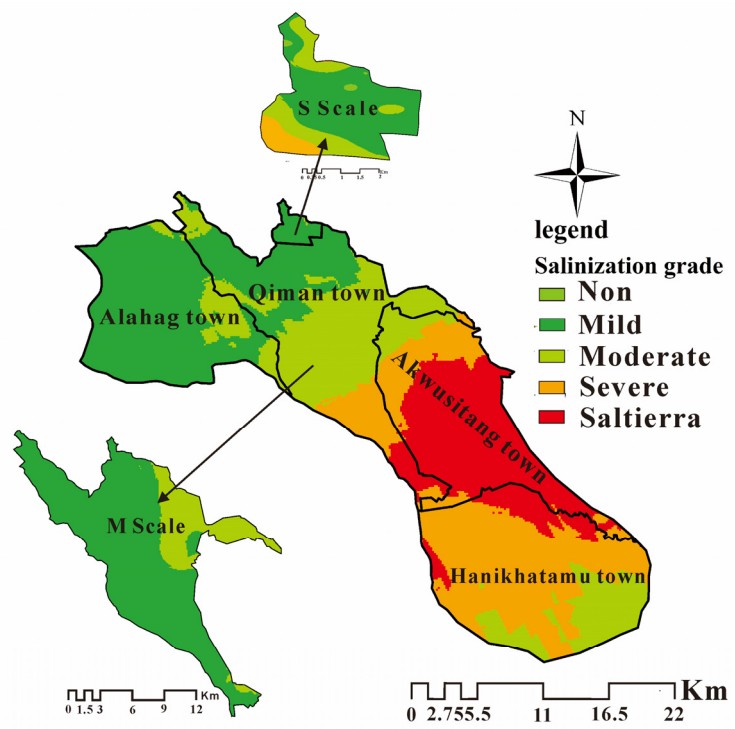

**Figure 8.** Spatial distribution of soil salinity at different scales.

**Table 5.** Area and proportion statistics of soil salt content at different scales.

| Soil Salt Content/% | Statistical Items | Scale | | |
|---|---|---|---|---|
| | | L | M | S |
| <0.1 (Non) | Area (km$^2$) | 0 | 0 | 0.09 |
| | Proportion (%) | 0 | 0 | 0.98 |
| 0.1–0.2 (Mild) | Area (km$^2$) | 200.37 | 161.60 | 5.61 |
| | Proportion (%) | 28.35 | 80.42 | 64.22 |
| 0.2–0.4 (Moderate) | Area (km$^2$) | 175.26 | 39.35 | 2.22 |
| | Proportion (%) | 24.79 | 19.58 | 25.40 |
| 0.4–0.6 (Severe) | Area (km$^2$) | 193.05 | 0 | 0.82 |
| | Proportion (%) | 27.31 | 0 | 9.39 |
| >0.6 (Saltierra) | Area (km$^2$) | 138.16 | 0 | 0 |
| | Proportion (%) | 19.55 | 0 | 0 |

According to Figure 8 and Table 5, on the L scale, the percentage of mild, moderate and severe salinization did not vary much, in which, the area of mild salinization was 200.37 km$^2$, accounting for 28.35% of the total area, distributed in the northwest of the study area. The moderate salinization accounted for 24.79% of the total area, and most of it was distributed in the middle of the study area. Severe salinization accounted for 27.31% of the total area. The saltierra accounted for 19.55%, which was mainly distributed in the southeast of the study area.

On the M scale, the soil salt content was between 0.123–0.344, mainly mild and moderate salinization, of which, the maximum area of mild salinization was 161.60 km$^2$,

accounting for 80.42%, and the rest was moderate salinization, accounting for 19.58%. Although the degree of salinization was low, the increasing direction was consistent with the flow direction of the Yingdaya River.

On the S scale, the spatial distribution regularity of the soil salinity was not strong, in which, non-salinization was distributed sporadically, accounting for 0.98%. Dabozi Village was mainly planted with wheat, and, during the sampling period, this section underwent irrigation. After irrigation, the salt in the farming layer moved to the deep soil with water. Therefore, the salinization in the farming layer was mainly mild, accounting for 64.22%. Moderate and severe salinization were distributed in strips in the north and south Dabozi Village, accounting for 25.4% and 9.39%, respectively.

### 3.3.2. Scale Effect and Multi-Scale Nesting of Soil Salinity Spatial Distribution

(1) Scale effect of the spatial distributions of soil salinity

Different sampling scales can only characterize corresponding spatial structure characteristics and change rules. By comparing the spatial distribution characteristics of soil salt at the three scales (Figure 7), the spatial distribution structure of the soil salt also changes with the increase or decrease in the sampling scales. At the S scale, the soil salinity shows non-salinization, as well as mild, moderate and severe salinization, and the S scale in the M and L scales shows singular mild salinization. At the M scale, the soil salinity shows mild and moderate salinization, and, on the M scale in the L scale, there are many types of salinization (mild, moderate, severe, saline soil). The above phenomena indicate that the possible reason for this is that, with changes in the sampling scale, small-scale spatial structure features are easily covered by larger scales, and the variations under the larger scales are ignored because the sampling scales are too small [11,34].

(2) Multi-scale nesting and accuracy test of soil salt spatial structure

According to the above analysis, the soil salinity variation exhibited an obvious scale effect; even under the same scale conditions, the soil salinity shows directional characteristics. Therefore, it is necessary to construct a comprehensive semi-variogram to quantitatively summarize all the effective structural information to characterize the main characteristics of regionalized variables in irrigation areas. The main method of structural analysis is the nested structure [35,36]. Implementation method: initially, the soil salt content at L, M and S scales under anisotropic conditions were nested in different directions, which were converted into isotropy, and then all of the converted isotropic structures were superimposed to form a unified nested model.

The spatial variation structure of soil salinity at L, M and S scales was anisotropic nested, and the steps are as follows:

(a) Soil salt anisotropic nesting at L scale:

$$\gamma_2(h) = \begin{cases} 0 & h = 0 \\ 0.794 + 1.851\left[\frac{8.244}{2}\left(\frac{h}{35430}\right) - \frac{6.341}{2}\left(\frac{h}{35430}\right)^3\right] & 0 < h < 35430 \\ 2.654 & h > 34540 \end{cases} \quad (9)$$

(b) Soil salt anisotropic nesting at M scale:

$$\gamma_1(h) = \begin{cases} 0 & h = 0 \\ 0.0164 + 0.636\left(2 - e^{-\frac{h}{16910}} - e^{-\frac{2.975h}{16910}}\right) & h > 0 \end{cases} \quad (10)$$

(c) Soil salt anisotropic nesting at S scale:

$$\gamma_0(h) = \begin{cases} 0 & h = 0 \\ 0.223 + 0.339\left[\frac{9.798}{2}\left(\frac{h}{3460}\right) - \frac{12.635}{2}\left(\frac{h}{3460}\right)^3\right] & 3460 > h > 0 \\ 0.562 & h > 3460 \end{cases} \quad (11)$$

According to the geostatistical scale nested theory, the nugget in the nested model is the variability that the smallest scale (S) cannot represent, i.e., 0.223, and the partial sill (C) is the sum of the partial sill on each scale. According to the above method, the final nested model of L, M and S is shown in Equation (12).

$$\gamma(h) = \gamma_0(h) + \gamma_1(h) + \gamma_2(h) \tag{12}$$

$$\begin{cases} 0.233 & h > 0 \\ 0.223 + 0.636\left(2 - e^{-\frac{h}{16910}} - e^{-\frac{2.957h}{16910}}\right) + \frac{18.573}{2}\left(\frac{h}{35430} + \frac{h}{3460}\right) - \frac{16.02}{2}\left(\frac{h^3}{35430^3} + \frac{h^3}{3460^3}\right) & 3460 > h > 0 \\ 0.562 + 0.636\left(2 - e^{-\frac{h}{16910}} - e^{-\frac{2.957h}{16910}}\right) + 1.851\left[\frac{8.244}{2}\left(\frac{h}{35430}\right) - \frac{6.341}{2}\left(\frac{h}{35430}\right)^3\right] & 35630 > h > 3460 \\ 2.654 + 0.636 & h > 35430 \end{cases} \tag{13}$$

After fitting the semi-variogram model at the three scales of irrigation area, township and village through the nested structure model, the Kriging interpolation method was used to perform spatial estimation on the sample data of the three scales, and the interpolation results are shown in Figure 9. To compare the prediction accuracy of the single-scale (different sampling density) ordinary Kriging interpolation method with that of the multiscale nested model method, the optimality of the theoretical semi-variogram model was tested using the I-value test and the cross-validation method. It can be seen from Table 6 that the *I* values of the multi-scale nested model method and $\overline{[\hat{Z}(x_i) - Z(x_i)]^2}$ were both smaller than the single-scale ordinary Kriging interpolation method, which indicates that the multi-scale nested model method exhibited a better expressive power for the spatial variation structure of soil salinity, and could effectively reveal the spatial variation law of the soil salinity.

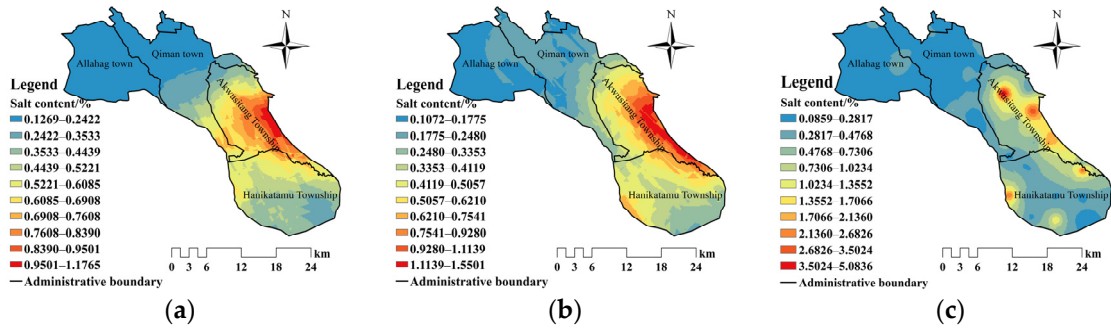

**Figure 9.** Spatial distribution of soil salt content: single-scale ordinary Kriging interpolation method for low-density sampling (**a**); single-scale ordinary Kriging interpolation for high-density sampling (**b**); multiscale nested model interpolation (**c**).

**Table 6.** Comparison of interpolation accuracy in different spaces.

| Interpolation Method | *I* Value | $\overline{[\hat{Z}(x_i) - Z(x_i)]^2}$ |
|---|---|---|
| Single-scale ordinary Kriging (q) | 15.692 | 15.275 |
| Single-scale ordinary Kriging (Q) | 11.861 | 11.334 |
| Nested model method | 6.797 | 6.769 |

Note: q represents 44 samples, Q represents 131 samples

Comparing the nested model interpolation map with the single-scale ordinary Kriging interpolation map (Figure 8), the overall distribution law is basically the same, and the salt content gradually increases from northwest to southeast, reflecting the large-scale structural characteristics. The structural factors of the same townships are similar. At the same time, the study area implements an intensive farmland management mode with the township as the basic unit. Therefore, the crop types and cultivation management modes

of each township are basically the same, resulting in uniform changes in the soil salt in all directions at the township scale. It can be seen in Figure 9a,b that the soil salt in each township is distributed in irregular spots, and the changes are uneven. Figure 9c shows that the soil salinity in each township is relatively balanced, especially in Akwusitang Township, and that it weakens in a wavy small range centered on the high-value areas, reflecting its sensitivity to small-scale distances. The overall change ranges of the soil salinity under the multi-scale nested model are much larger than those of single-scale ordinary Kriging interpolation. Specifically, the estimation of the high-value area in the southeast of the study area is more in line with the actual situation, which shows that the nested model method not only takes into account large-scale structural factors, but also takes into account small-scale local variation characteristics.

## 4. Discussion

The soil salinity in the study area is affected by various factors on different scales to form a scale effect [37]. These factors include meteorological factors, geoscience comprehensive factors and land management factors. The three types of factors are coupled and jointly act on the soil salinization process. Among them, the landform lays the foundation for the formation of salinization, the extreme arid climate environment and land use mode affect the development trend of salinization and the groundwater environment directly affects the formation of salinization [38]. In order to determine the inter-relationships and action intensities between various influencing factors and soil salinization at different scales in the study area, this section provides a quantitative analysis on the influencing factors of salinization through the gray correlation degree. As the overall climate changes in the study area were not large, climate factors were not considered in the quantitative analysis. Groundwater depth, groundwater mineralization, slope land, elevation, soil texture and land use mode were selected as the impact data series, and were extracted with their respective spatial variables (Figure 10). The soluble salt contents of the sampling points were taken as the reference sequences, which came from the indoor analysis results.

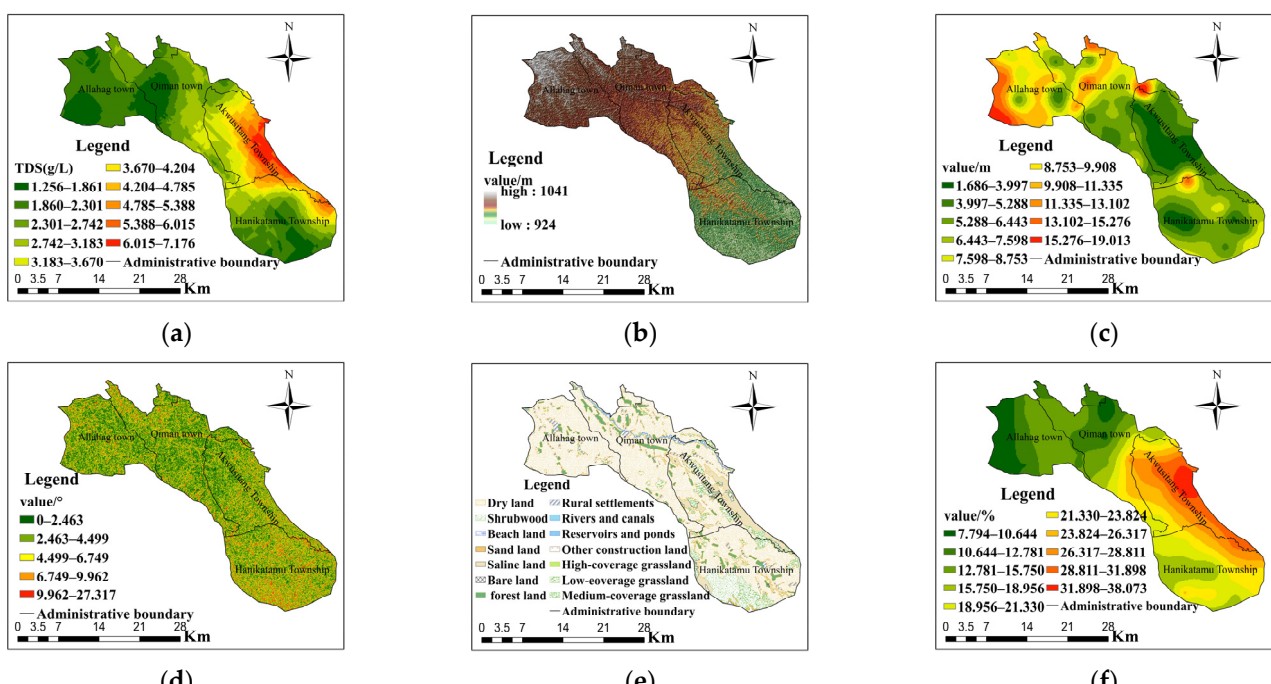

**Figure 10.** Spatial distribution data of the influence factors of salinization: groundwater mineralization (TDS) (**a**); surface elevation (**b**); groundwater burial depth (**c**); topographic slope (**d**); land use type (LUP) (**e**); silt content (**f**).

We calculated the degree of correlation between six types of influencing factors and soil salt content at different scales according to Formulas (7) and (8), as shown in Table 7. In a grey system, the higher the grey relational coefficient, the greater the importance of the factor to salinization formation and evolution; as shown in Table 7, the six influence factors affected soil salinity at different scales in a significantly different manner. In the L scale, the factors were ranked in descending order of the grey relational coefficient, with the soil salt content as follows: groundwater mineralization > silt content > groundwater burial depth > elevation > land use type > slope; in the M scale: groundwater mineralization > silt content > groundwater burial depth > elevation > land use type > slope; and, in the S scale: slope > groundwater burial depth > groundwater mineralization > silt content > land use type > elevation.

**Table 7.** Correlations between the influence factors of salinization at different scales.

| Scale | Subsequence | | | | | |
|---|---|---|---|---|---|---|
| | Elevation | Groundwater Mineralization | Groundwater Burial Depth | Slope | Land Use Type | Silt Content |
| L | 0.507 | 0.756 | 0.635 | 0.327 | 0.422 | 0.721 |
| M | 0.409 | 0.589 | 0.427 | 0.342 | 0.402 | 0.521 |
| S | 0.201 | 0.382 | 0.391 | 0.541 | 0.218 | 0.353 |

In a grey system, the greater the correlation degree, the stronger the leading role that it plays in the evolution of salinization. It can be seen from Table 7 that, with decreases in the scale, the correlation degrees between other influencing factors and the soil salt content show a gradually decreasing trend in general, except for the slope. On the L-scale, the groundwater mineralization, soil texture, groundwater burial depth and elevation correlation are all greater than 0.5, ranking in at the forefront, indicating that these four influencing factors are the main factors affecting soil salinization on that scale. Changes in elevation also directly affect the flow of the groundwater and surface water, which also affects the movement and accumulation of soil salt [39]. From the perspective of large terrain, water-soluble salts migrate from high to low with the water and accumulate in low-lying areas [40]. Figure 10 shows that the northwestern region of the study area featured high terrain, deep groundwater and low mineralization, so the salt there was mainly discharged. As the surface and groundwater migrated to the low-lying southeast, the groundwater depth gradually became shallower, and the mineralization degree became larger. Finally, strong evaporation leads to serious salinization in the southeast. In the southeastern part of the study area, the silt content was high, the soil texture was light and the impact of the soil texture conditions on salinization was reflected in the water conductivity. The capillary pores of silty sand were moderate, and the water conductivity was strong. The groundwater flows through the capillaries and rises at high heights and fast speeds, which is conducive to the formation of salt deposits on the surface [19,41].

On the S scale, the correlation between the soil salt content and the slope was greater than other influencing factors. The irrigation area was located in the middle and lower part of the alluvial–proluvial fan, so the elevation change range within the S scale was small, but the original terrain was changed during the process of the land preparation, cultivation and excavation of the canal system, which made the micro terrain slope change greatly at small ranges and weakened the impact of elevation on soil salinization [42,43]. In the micro terrains, as it was affected by the slope after irrigation, the water containing salt flowed from high elevations to lower elevations. As the water evaporated, the salt stayed in the lower-lying areas. Therefore, on the S scale, the impact of the slope on the soil salt was greater than the elevation. On the L scale, there were various types of land uses, including saline alkali land, grasslands, bushes, arable land, etc. Under different land use modes, soil physical and chemical properties, surface evaporation intensities (surface temperature) and vegetation physiological characteristics can all be different, which causes uneven distributions of water levels and water quality in space and time [44,45], and

inevitably has different effects on soil salinization at the L scale. On the S scale, the land use mode is singular, so the correlation between the soil salt content and the land use type is obvious lower than for the L scale and M scale.

Traditional flood irrigation was the main irrigation method in the study area, so irrigation and drainage have significant impacts on the soil salt content in the irrigation area. The Yingdaya River runs through Qiman Town, and irrigation and alkali drainage channels of all sizes are distributed around the river, as shown in Figure 11a. Therefore, taking the Yindaya River as the research object, 25 points were selected from upstream to downstream in the river, and the sample points were evenly distributed on both of its banks. According to the results for the sample points and Kriging interpolation, a salt content distribution profile for the 0–30 cm soil layer was drawn, and the soil salinization along the river direction at the M scale was analyzed. Figure 11b is the profile of the total salt content of the Yingdaya River. It can be seen from the figure that the soil salt content increases with the trends of the river, showing a lower distribution upstream and a higher distribution downstream. At the same time, it can also be seen from the figure that some alkali drainage channels are connected with the Yindaya River, which inevitably discharges the soil salt from the irrigation area directly into the Yindaya River through the irrigation channels, thus further aggravating the degree of salinization downstream in the river.

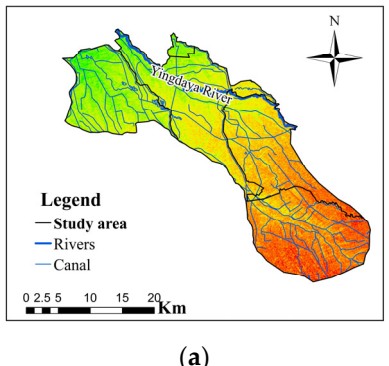
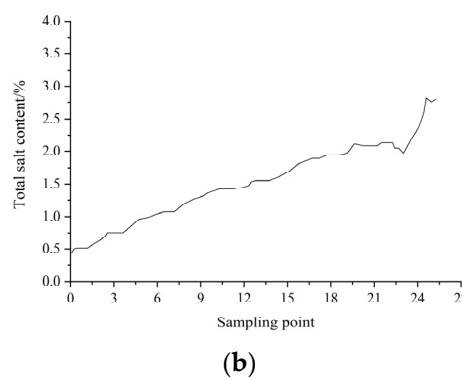

(**a**)                    (**b**)

**Figure 11.** Distribution profile of salt content in 0–30 cm soil layer of water system and Yingdaya River in the study area: water system map of study area (**a**); profile map of salt content distribution in 0–30 cm soil layer of Yindaya River (**b**).

## 5. Conclusions

(1) Influenced by different-scale structural factors and random factors, the choice of small and medium-scale study sites causes soil salinity to tend to be homogeneous on small scales and heterogeneous and complex on large scales. The soil salinity was ranked as: L scale (1.664%) > M scale (0.263%) > S scale (0.217%), and the coefficient of variation was ranked as: L scale (2.564) > M scale (1.312) > S scale (0.866).

(2) Under isotropic conditions, the nugget coefficients of L, M and S soil salinity are 0.298, 0.387 and 0.613, respectively, with ranges of 21.56 km, 13.31 km and 1.685 km, all of which are in accordance with medium spatial correlation. As the scale increases, the spatial autocorrelation of soil salinity increases and the nugget effect decreases. Under anisotropic conditions, the direction of maximum variation on the L and M scales was northwest-southeast, and the direction of maximum variation on the S scale was east-west (tillage direction).

(3) From the values of the *I*-value test and cross-test, the accuracy of the estimation of the multi-scale nesting model method is better than that of the estimation of single-scale ordinary Kriging. In terms of distribution characteristics, the spatial distribution of soil salinity fitted by the ordinary Kriging method and the nesting model method is the same in the overall trend, but the nesting model method is clearer and more comprehensive in expressing local characteristics of the soil salinity spatial variation.

(4) In the L scale, the factors were ranked in descending order of the grey relational coefficient, with the soil salt content as follows: groundwater mineralization > silt content > groundwater burial depth > elevation > land use type > slope; in the M scale: groundwater mineralization > silt content > groundwater burial depth > elevation > land use type > slope; and in the S scale: slope > groundwater burial depth > groundwater mineralization > silt content > land use type > elevation.

**Author Contributions:** Conceptualization, S.L. and L.L.; methodology, S.L. and R.W.; software, L.L.; validation, D.S. and R.W.; formal analysis, D.S. and R.W.; investigation, D.S. and L.L.; re-sources, L.L.; data curation, L.L., R.W. and S.L.; writing—original draft preparation, L.L.; writing—review and editing, S.L.; visualization, D.S. and R.W.; supervision, L.L.; project administration, R.W.; funding acquisition, S.L. All authors have read and agreed to the published version of the manuscript.

**Funding:** This research was funded by the National Natural Science Foundation of China, grant number U1603243; Natural Science Foundation of Xinjiang Uygur Autonomous Region, grant number 2022D01C40.

**Data Availability Statement:** Some data used during the study are available from the author by request (email: luli0401@sina.com (L.L.)) and through (http://www.gscloud.cn accessed on 19 February 2021) download.

**Conflicts of Interest:** The authors declare no conflict of interest.

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
