# Peer review of "Study on the Scale Effect of Spatial Variation in Soil Salinity Based on Geostatistics: A Case Study of Yingdaya River Irrigation Area"

_land, doi:10.3390/land11101697_

Round 1

Reviewer 1 Report

General comments

The topic of the manuscript is interesting, but the writing and methodology is poor and messy. This needs significant improvement. Especially the Introduction and Discussion need improvements. The English is poor and should be revised thoroughly by a native speaker.

‘The’ and some other words are often used unnecessarily. Check the entire document and remove words where needed. This makes the text very hard to read.  

I’d suggest the authors to read the manuscripts of Alexandre Wadoux, who spend a PhD thesis on sampling optimization.

Introduction

In general the introduction is weak. Literature is not well linked to the aim of the study. The aim is not mentioned explicitly, which makes it a ‘nice exercise’, but I don’t understand why this study should be published. The added value is missing. However, I think this manuscript has potential and therefore I’d like to recommend ‘major revisions’.

L 2: before "A case study" write : or ;

L 20: L, M and S are not yet defined.

L 37: "as research has deepened": I don't understand what is meant by deepened. Please rephrase.

L39: "the concept of the scale" I don't understand this sentence. Please clarify.

L 39-41: this is absolutely not true! There are many scale-related rules for pedon description and soil mapping. In traditional soil mapping it was more clear than after the introduction of Digital Soil Mapping.

L42-45: why did problems with scale become increasingly important after 1990? I don't understand "a large nummer of scientists explored the scale effect of soil"? Effect on what?

L52. Mention the three cases briefly before discussing them in detail by referring to different studies. Also mention how these three cases are linked to your study. This is unclear to me at the moment.   

L53. Year after author is missing (check whole manuscript)

L55. What is a ‘spatial variation law’?

L73. ‘the present study’, do you mean your study? If yes, please rephrase by saying ‘this study’.

L79. I think the authors mean ‘spatial variation’ instead of ‘spatial variation structure’

L79. It is ‘at different scale levels’ and not ‘in different scales by’.

L81. Remove ‘characteristics’.

L83. I don’t understand this sentence, so please rephrase. A reference and basis of what? ‘Optimizing sampling solutions’, which problems need to be solved? ‘Spatial management level’, what is this? Is this one of the scales you are looking at? Is this the aim of the study?

Materials and Methods

Why did you choose to collect the soil samples in a grid design? In geostatistics variable distances to a sampling point is needed for best spatial interpolation. Please clarify this in the text as well, because this is a crucial element of the study.   

I miss information on the soil variability and soil types in the region! This information is essential to make a link with salinity level.

L85. Chapter 2 has the same header as 2.1, while Chapter 2 needs to have the same header as Chapter 1.

L96. This sentence is not starting with captial letter, ‘crisscrossing’ is not approriate in this context, so use braided or meandering rivers, and it is an unfinished sentence.

L103. Mention what we see from a until f. If these figures represent L, M and S, please introduce this before 2.2.

L104. What depth are the soil samples taken?

L105. A finite verb is missing. Also explain what the GIS fishing net tool and the grid nesting method is.

L120. Were they stored in a fridge as well? How much time passed between the sampling and the analysis? Does this influence your results?

L128. How long did you shake? Which filter did you use? Please add more detail and remove the word ‘briefly’ in L128.  

L134. Were you able to create a semi-variogram while the distance between the sampling points is equal?

L162. This sentence starts not with a capital letter. Please correct this sentence.

 Results

L201. Remove ‘and analysis’. It is either ‘Results and discussion’ or just ‘results’.

L205. What is ‘the traditional statistical level’? Is there also a non-traditional statistical level?

L209. I’m not very familiar with ‘salt content of soil’. Is this something different than the Cation Exchange Capacity?

L227. Mention which structural and random factors you think of. Is diversity in soil properties (e.g., texture) and soil type not clarifying most of the variation?

L239-L241. This is a strange sentence, please rephrase.

L244. Again, this sentence does not start with a capital letter. Please correct.

L258. Please be consistent in the ‘scale’ you are referring to. ‘research scale’ is not a scale.

L284-286. In the methodology the use of environmental variables is missing. Include this in the methodology and add a table in the annex with the main characteristics, scale and source of the explanatory variables used.

L304-307. Is this not because ‘river flow direction’ is an explanatory variable that turns out to have large influence on the model fit? It is interesting to show which environmental variables were selected at S, M and L level, because these probably differ much. Explore why these differ.

L328. This is a very simple method. You should consider using regression kriging at least (and include environmental (explanatory) variables to explain the spatial variation).

Figure 7. Use the same format as Figure 2 to make it clearer. Also consider using the same scale in all three figures to be able to compare the variation captured at different scales.

Discussion

General comment: re-write this section. Put it in a broader perspective and underpin the statements correctly. This is currently not done.

L403-404. Be more specific than ‘different factors at different scales’. For me Figure 8 shows that some soil samples showed high values and some showed low values and nothing else but the soil analyses caused the spatial variation.

L411. Do not refer to yourself. Besides, this sentence is incorrect.

L412-414. What do you mean by ‘best scale’ and what by ‘high precision estimation’ can reduce the workload of sampling and analysis? Can the number of samples be reduced? I don’t understand this sentence.

Reviewer 2 Report

Manuscript #Land-1872279:

This article addresses an important and classical problem: the scale effect of the spatial variation of soil characteristics. It analyzed the scale effect of soil salinity estimates and variation characteristics in different spatial ranges in the Yingdaya River irrigation area and provided an improved multi-scale nested Kriging interpolation model, which provides ideas for spatial variation analysis of soil physical and chemical properties. The research ideas in this article have contributed to the development of spatial interpolation model research.

In my suggestion, if this article contains the comparative analysis results of interpolation models based on different sampling densities at the same spatial scale. It will be more complete. Another suggestion is that this paper needs a fuller discussion.

     -------------------------------Minor comments----------------------------------- 

1. Language problem:

I understand that English is not the authors' native language. It is recommended that the authors seek assistance to improve writing.

P.2 Line 92: “Hanikatamu township” should rewrite as “Hanikatamu Township”.

P.2 Line 95: “in summer. The” rather than “in summer, the”.

P.2 Line 96: “crisscrossing rivers” should be “There are crisscrossing rivers”.

P.2 Line 97: “cotten, Yingdaya” should be “cotton. The Yingdaya”.

P.3 Line 98: “in the territory, due to” should be “in the territory. Due to”.

P.3 Line 100-101: “The location of the study area is shown Figure 1.” should rewrite as “The location of the study area is shown in Figure 1.”.

P.3 Line 105: Missing subject.

P.3 Line 111: “is 4.4 km”, not “is 4.4km”.

P.3 Line 112: “town. The”, not “town, the”.

P.3 Line 118: “mineralization”, not “Mineralization”.

P.4 Line 137: “For”, not “for”.

P.7 Line 210: “0.217%, respectively”, rather than “0.217% respectively”.

P.7 Line 231: “the overall changing characteristics”, rather than “the overall change characteristics”.

P.7 Line 244: “The nugget (C0)”, rather than “nugget (C0)”.

P.7 Line 246: “soil salinity were 0.613, 0.387, and 0.298, respectively”, rather than “soil salinity are 0.613, 0.387, and 0.298, respectively”.

P.8 Line 265-266: The words are incorrectly described. The soil salinity ranges (A) are 1.685, 13.310, and 21.46 km, respectively, corresponding to the model parameters of the three scales. Not range.

P.8 Line 271: “became”, rather than “becomes”.

P.8 Line 277-279: Please check the singular and plural of the noun.

P.9 Line 296: “northwest-southeast. This was”, rather than “northwest-southeast, this was”.

P.10 Line 335-336: “and most of it was”, rather than “and most of them were”.

P.10 Line 351: “water. Therefore”, rather than “water, therefore”.

P.11 Line 352: “were”, rather than “was”.

2. Tables and Figures:

The figures and tables could be improved and perhaps consolidated, e.g. Fig. 7 and Fig. 8. The methods, e.g., 2.4 Data Analysis and Methods, could be shortened.

The map of China's political districts in Figure 1a is not standardized.

Please carefully check the scale and font format in each figure's uniformity. The fonts in many figures are not clearly displayed.

It is recommended to add a boundary line like Figure 8 to Figure 7.

Reviewer 3 Report

See attached file

Round 2

Reviewer 3 Report

I went through the revised version of the manuscript and found that it had considerably improved from the original manuscript. The language and structure of the manuscript have improved and are more understandable. Therefore, the article is suitable for publication in its present form.